# Screening for Influenza and Morbillivirus in Seals and Porpoises in the Baltic and North Sea

**DOI:** 10.3390/pathogens12030357

**Published:** 2023-02-21

**Authors:** Iben Stokholm, Christine Baechlein, Sara Persson, Anna Roos, Anders Galatius, Line Anker Kyhn, Signe Sveegaard, Charlotte Bie Thøstesen, Morten Tange Olsen, Paul Becher, Ursula Siebert

**Affiliations:** 1Section for Molecular Ecology and Evolution, GLOBE Institute, University of Copenhagen, Øster Farimagsgade 5, 1353 Copenhagen, Denmark; 2Institute for Terrestrial and Aquatic Wildlife Research, University of Veterinary Medicine Hannover, Werftstr 6, 25761 Büsum, Germany; 3Institute of Virology, University of Veterinary Medicine Hannover, Bünteweg 17, 30559 Hannover, Germany; 4Lower Saxony State Office for Consumer Protection and Food Safety, Food and Veterinary Institute Braunschweig/Hannover, Eintrachtweg 17, 30173 Hannover, Germany; 5Swedish Museum of Natural History, Department of Environmental Research and Monitoring, Frescativägen 40, SE-104 05 Stockholm, Sweden; 6Marine Mammal Research, Department of Ecoscience, Aarhus University, Frederiksborgsvej 399, 4000 Roskilde, Denmark; 7Fisheries and Maritime Museum, Tarphagevej 2, 6710 Esbjerg, Denmark

**Keywords:** influenza virus, morbillivirus, pinnipeds, harbour seal, grey seal, ringed seal, cetaceans, harbour porpoise, marine mammals, pathogen screening

## Abstract

Historically, the seals and harbour porpoises of the Baltic Sea and North Sea have been subjected to hunting, chemical pollutants and repeated mass mortalities, leading to significant population fluctuations. Despite the conservation implications and the zoonotic potential associated with viral disease outbreaks in wildlife, limited information is available on the circulation of viral pathogens in Baltic Sea seals and harbour porpoises. Here, we investigated the presence of the influenza A virus (IAV), the phocine distemper virus (PDV) and the cetacean morbillivirus (CeMV) in tracheal swabs and lung tissue samples from 99 harbour seals, 126 grey seals, 73 ringed seals and 78 harbour porpoises collected in the Baltic Sea and North Sea between 2002–2019. Despite screening 376 marine mammals collected over nearly two decades, we only detected one case of PDV and two cases of IAV linked to the documented viral outbreaks in seals in 2002 and 2014, respectively. Although we find no evidence of PDV and IAV during intermediate years, reports of isolated cases of PDV in North Sea harbour seals and IAV (H5N8) in Baltic and North Sea grey seals suggest introductions of those pathogens within the sampling period. Thus, to aid future monitoring efforts we highlight the need for a standardized and continuous sample collection of swabs, tissue and blood samples across Baltic Sea countries.

## 1. Introduction

Marine organisms are under pressure from a broad range of environmental and anthropogenic stressors, including climate change, plastic, noise and chemical pollution and pathogen-induced mass mortalities [1,2,3,4,5,6]. This is also the case in the Baltic Sea and North Sea, where extensive hunting and exposure to persistent organic pollutants (POPs) led to dramatic declines and local extinctions among the region’s marine mammal populations in the 19th and 20th centuries [7,8,9,10]. The establishment of protective measures for marine mammals in the 1970s led to significant increases in the harbour seal (*Phoca vitulina*), grey seal (*Halichoerus grypus*) and ringed seal (*Pusa hispida*) populations [11,12,13]. Further, monitoring of the North Sea/Skagerrak and the Western Baltic (Kattegat, Belt Sea and the German western Baltic) harbour porpoise (*Phocoena phocoena*) populations, points to stability in their abundance and distribution [14]. However, the population of the Baltic Proper harbour porpoises is threatened with extinction, possibly due to bycatch, prey depletion, chemical and/or noise pollution [15,16,17,18].

Given their high trophic level, both the harbour porpoise and the three seal species are used as indicator species in biomonitoring, e.g., within the Helsinki Commission (HELCOM) marine monitoring framework. This includes the monitoring of infectious and zoonotic diseases in each of these key species [19]. Throughout the past four decades, the European harbour seal populations have been affected by four unusual mortality events (UMEs) all originating from an epicenter in Kattegat and spreading to multiple parts of the Baltic Sea–North Sea region [20,21,22,23]. Of these, three have been linked to infections with the highly contagious respiratory viruses phocine distemper virus (PDV), which belongs to the genus *Morbillivirus* (family *Paramyxoviridae*), and influenza A virus (IAV) (family *Orthomyxoviridae*). PDV first appeared in 1988 and re-emerged in 2002 resulting in two mass mortality events killing ~23,000 and ~30,000 seals, respectively [23]. Another UME occurred in 2014, where the transmission of IAV (H10N7) from birds to harbour seals resulted in the death of ~2000–2500 animals [24,25]. No information is currently publicly available on the presence of PDV and IAV among the vulnerable Baltic ringed seals.

Morbilliviruses and influenza are also found among cetaceans where infections with cetacean morbillivirus (CeMV) may occasionally develop into a central nervous system (CNS) form characterized by lesions in the brain [26]. Since the first discovery of CeMV in harbour porpoises collected along the North Irish coast in 1988 [27,28], an increasing number of CeMV strains have been linked to disease during mass mortalities, UMEs and sporadic deaths in multiple odontocetes and mysticetes worldwide [26,29,30,31,32,33]. In contrast to CeMV, influenza infections have only rarely been detected in cetaceans with few viral or serological cases in *Balaenopteridae* [34], pilot whales (*Globicephala melaena*) [35], beluga whales (*Delphinapterus leucas*) [36], Dall’s porpoises (*Phocoenoides dalli*) [37] and common minke whales (*Balaenoptera acutorostrata*) [37]. One study investigating the seroprevalence in harbour porpoises of the North Sea did not detect influenza A and B in this species [38]. Such knowledge on the general distribution and prevalence of IAV in cetaceans is currently limited.

Regular monitoring of pathogens is essential to the evaluation of population health, zoonotic risk assessments, disease circulation and the identification of natural reservoirs for zoonotic diseases. However, with a few exceptions, screenings for pathogens in Baltic Sea–North Sea seals and porpoises are often conducted on an opportunistic basis with logistical and economic constraints preventing the collection of high-quality samples from geographically and demographically representative parts of the populations [39]. The fluctuating sample size and limited species coverage lead to several spatiotemporal gaps in the screening efforts, raising questions as to what extent PDV, CeMV and IAV are circulating in the Baltic Sea and North Sea marine mammal populations. In the present study, we conducted the first large-scale survey of IAV, PDV and CeMV on archived and new samples collected from seals and porpoises in the Baltic Sea–North Sea region between 2002–2019 to investigate the occurrence of these pathogens across species, space, and time.

## 2. Materials and Methods

### 2.1. Sampling

Tracheal swabs and lung tissue samples were collected from harbour seals (99), grey seals (126), ringed seals (73) and harbour porpoises (78) in the period 2002–2019 along the Danish, German and Swedish coastlines (Table 1; Figure 1; Appendix A). These were collected during necropsies of animals that had been found stranded, bycaught or were subjected to regulatory hunt. Samples from necropsies were stored at −70 °C shortly after collection, whereas samples from animals necropsied in the field (Appendix A: PH-SE-057 to PH-SE-061, PH-SE-064 to PH-SE-073) were kept cold and transported on dry ice prior to long-term storage at −70 °C. Additional samples from the time period 2002–2017 were obtained from the archives at the Institute for Ecoscience (Aarhus University) and the Environmental Specimen Bank (ESB) at the Swedish Museum of Natural History with storage conditions at −20 °C. To increase the chances of detecting and sequencing PDV, CeMV, or IAV, preference was given to animals with signs of respiratory disease and a decay state of 1–3 out of 5 [40,41], which implies that the carcasses were relatively fresh.

### 2.2. Virus Screening

The viral RNA extractions were performed using the IndiMag Pathogen Kit w/o plastics (Cat.-No.: SP947257), with homogenization of the tissue samples in the BeadMill 24 (Thermo Fisher, Waltham, MA, USA) using the *Lysing Matrix* M (mpbio) and buffer RA1 + ß-Mercaptoethanol (Macherey/Nagel). All screenings were made by RT-qPCR with the generic morbillivirus primers MVP2202 (5′-KKC TCR TGGTWC CW R CAG GC-3′) and MVP2480R (5′-TCT CTY CTGTGC CCT TTT TAA TGG-3′) used in [42] and the IAV primers NP-1448-F 5′- GGGAGTCTTCGAGCTCTC-3′ and NP-1543-R 5′- GCATTGTCTCCGAAGAAATAAGA-3′ with probe IAV-NP-1473-FAM FAM-AAGGCAVCGARCCCGATCGTGC-TAMRA [43,44]. The morbillivirus RT-qPCR was performed with the QuantiTect SYBR Green RT-PCR Kit (Qiagen, Hilden, Germany). Each reaction consisted of 12.5 µL QuantiTect SYBR Green RT-PCR Mastermix (Qiagen), 1 µL forward primer (20 pmol) MVP2202, 1 µL reverse primer (20 pmol) MVP2480, 0.25 µL reverse transcriptase, H_2_O 5.25 µL and 5 µL RNA. Detection of influenza virus RNA was conducted using the QIAGEN QuantiTect Probe RT-PCR kit. Reactions were performed based on a mixture of 12.5 µL QuantiTect Probe RT-PCR mix, 2.0 µL primer and probe mix (NP-1448-F: 25 µL, NP-1543-R: 25 µL, NP-1473-FAM: 3 µL, H_2_O: 147 µL), 2 µL GFP primer and probe (HEX) mix (for internal control), 0.25 RT-Mix, 3.25 µL H_2_O and 5 µL sample. RNAs extracted from supernatants of cells infected with either PDV or IAV (H1N1) were used as positive controls in the two assays, respectively. For both assays cycling conditions were set to 50 °C at 30 min, 95 °C at 15 min, 40 cycles of 30 s at 95 °C, 30 s at 56 °C and 30 s at 72 °C. To decrease processing time, viral screenings were performed on extractions of pooled samples of 3–5 individuals or on pools of 3–5 extracts on individual samples (Appendix A).

## 3. Results & Discussion

### 3.1. Viral Pathogens Only Detected in Samples Collected during Known UMEs

In this study, a total of 298 pinnipeds including 47 seal pups, 77 juveniles, 105 adults and 69 of unknown age and 78 harbour porpoises including 9 calfs, 32 juveniles, 11 adults and 26 of unknown age were screened for PDV, CeMV and IAV (Table 2). Of the 376 marine mammals, only 3 seals tested positive; a male harbour seal pup collected on 13 September 2014 in Kattegat (Sample number: PV-DK-027, Ct: 32.30) and a male harbour seal of unknown age collected on 29 August 2014 in the Limfjord (Sample number: PV-DK-030, Ct: 28.45) tested positive for IAV (Figure 1B; Appendix A). The tissue samples originated from 2 of 15 individuals collected during a known outbreak of IAV in 2014. In addition, lung tissue from an adult male harbour seal (Sample number: PV-SE-002, Ct: 25.42) collected in Storön, Sweden, in July 2002 tested positive for PDV (Figure 1B; Appendix A). This seal represents one of two individuals collected during the known PDV outbreak in 2002. All grey seals, ringed seals and harbour porpoises tested negative for IAV, CeMV and PDV. Thus, PDV and IAV were only detected in harbour seals collected during known UMEs.

### 3.2. Morbillivirus in North European Marine Mammals

The absence of PDV among Baltic Sea and North Sea seals outside of UMEs found in this study, corresponds well with previous serological investigations, which evidenced a decreasing prevalence of PDV-specific antibodies among harbour seals in the region following each of the UMEs in 1988 and 2002 [45]. These findings are supported by a recent molecular analysis indicating that the 2002 UME did not arise as a re-introduction of the 1988 strain from a reservoir species in northern Europe [21]. For now, the sparse contact between North European and Arctic pinniped populations seems to protect the harbour, grey and ringed seals inhabiting the North and Baltic Sea from continuous exposure to the enzootic circulation of Arctic PDV strains. However, the occasional detection of PDV in harbour seals stranded along the coast of Belgium and France in 1998 [46] and the Netherlands in 2014 (Genbank accession: KU342688) indicates repetitive introductions of PDV to seal populations in northern Europe, possibly from Arctic reservoirs such as harp seals, ringed seals and bearded seals [47,48,49,50,51,52,53]. Despite the low antibody prevalence and current high population density of harbour seals, none of these sporadic introductions have resulted in a UME. As the previous outbreaks of PDV in North European harbour seals both arose from an epicenter in central Kattegat between April and May [21,23,24], certain circumstances (e.g., place and timing) may be prerequisite for an introduction to initiate an outbreak in these populations.

The absence of CeMV in the screened harbour porpoises are in line with more recent studies finding no CeMV RNA or antigens among harbour porpoises collected in the Baltic Sea and North Sea between 1990–2016 [54,55]. However, evidence of CeMV infections in a white-beaked dolphin (*Lagenorhynchus albirostris*) on the North Friesian coast of Germany in 2007 [56] and a fin whale (*Balaenoptera physalus*) in Denmark in 2016 [33], suggests that occasional introductions of CeMV to the Baltic Sea–North Sea region occur. Further, findings of the porpoise morbillivirus (PMV) strain among North Sea harbour porpoises collected between 1987–1988 [27,28], and dolphin morbillivirus (DMV) strain specific antibodies in Baltic and North Sea harbour porpoises, common dolphins (*Delphinus delphis*), a long-finned pilot whale (*Globicephala melas*) and a white-beaked dolphin collected between 1991–1997 suggest widespread infections in this period [57,58,59]. While CeMV was first detected in harbour porpoises in northern Europe, subsequent studies of archival material identified CeMV-associated UMEs in northwest Atlantic bottlenose dolphins (*Tursiops truncatus*) as early as 1982 [60] as well as in the period between 1987–1988 [61]. These and more recent findings during the past decades have revealed a worldwide distribution of CeMV, with different strains circulating in a broad range of species and oceans [26,29,30,31,32,33,62]. Thus, while this study found no evidence of PDV and CeMV in pinnipeds and harbour porpoises collected from the partially enclosed region, the occasional introductions from migrating North Atlantic pinnipeds or cetaceans could seed a new epizootic with high impacts on the population’s health.

### 3.3. Influenza (IAV) in North European Marine Mammals

We detected IAV in two harbour seals collected in 2014, corresponding with the outbreak of H10N7 among North European pinnipeds [24]. Although we did not find positive cases outside of the 2014 UME, the recent find of H5N8 clade 2.3.4.4b infected Baltic grey seals in 2016–2017 [43] and a grey seal infected with H5N8 clade 2.3.4.4b collected in Sweden in 2021 [63] provides evidence of occasional spillover from avian reservoirs in the region. Recent spillovers have also been detected among North Sea seals with finds of a H3N8 infected grey seal collected on the coast of Cornwall in 2017 [64], H5N8 clade 2.3.4.4b found in four harbour seals and one grey seal in the UK in late 2020 [65], H5N8 clade 2.3.4.4b found in one harbour seal collected on Fyn, Denmark [63,66] and H5N8 2.3.4.4b found in harbour seals collected in 2021 in the German part of the Wadden Sea [44]. In addition, the detection of H4 antigens in 3 of 757 seals collected in the North Sea in 1988 [67] and positive NP-ELISA and hemagglutination inhibition tests of grey seals and harbour seals collected in 2011–2013 provide evidence of IAV infections among North Sea pinnipeds (2011 = 1:29, 2013 = 2:25) prior to the outbreak in 2014 [68]. Thus, North and Baltic Sea harbour and grey seals are subject to occasional exposure to IAV. However, further studies are required to elucidate the role of pinnipeds as potential mammalian reservoirs [69].

In contrast to seals, studies of influenza in cetaceans are sparse with detections of IAV-specific antibodies or viruses in; several individuals of *Balaenopteridae* in the South Pacific collected between 1975–1976 [34], one pilot whale in North America collected in 1984 (H13N2 and H13N9) [35], and seven common minke whales and two Dall’s porpoises (*Phocoenoides dalli*) in the western North Pacific collected between 2000–2001 [37]. Previous serological investigations conducted on harbour porpoises in the North Sea found no evidence of influenza A and B in 79 animals collected along the Dutch coast between 2003 and 2013, suggesting that influenza infections are uncommon in this species [38]. As such, our negative findings in harbour porpoises are consistent with previous findings in the North Sea.

### 3.4. Implications and Considerations for Future Screening Efforts

The present study represents the largest RT-qPCR-based screening of IAV, PDV and CeMV conducted on marine mammals in the Baltic Sea–North Sea region. As these screenings are limited to the detection of viral RNA, our results only represent acute infections with presence of viral nucleic acid among the animals at the time of death. In addition, false negatives may arise due to rapid degradation of RNA when exposed to UV-light, RNases and freeze-thawing. We accounted for this by: (i) prioritizing samples collected from animals with symptoms of respiratory infection and a decomposition state no more than 3 out of 5; (ii) focusing on samples collected from the trachea (swabs), lung and spleen (tissue), which are known to hold higher loads of virus in animals infected with influenza and morbillivirus. However, while the results do not indicate the presence of IAV, PDV and CeMV outside of epidemics, we cannot reject the possibility of occasional introductions of those viruses to seals or cetaceans in the area as indicated by the studies highlighted above.

Although the study represents the largest survey conducted in this region, the sample size is limited through time and space. In comparison to the current study, investigations of IAV prevalence among North American grey seals (345 pups and 57 adults) collected between 2013–2015 detected viral RNA in 5–12 % of the animals with a seroprevalence of 19% in pups and 50% in adults [69]. With a sample set of 402 individuals collected within a 3-year time span, this data set is significantly larger than the current one, which covers 298 pinnipeds (47 seal pups, 77 juveniles, 105 adults and 69 seals) spread across three species collected within a 17-year time span. Interestingly, no evidence of pathogenicity was detected among the American IAV-infected adult grey seals, and although acute infections were observed in all age groups the largest fraction was represented by recently weaned pups [69]. Thus, our negative results could be an artifact of a sample set limited by its size and age-associated susceptibility towards IAV.

Since the protection of marine mammals in the 1970s, the Northeast Atlantic pinniped populations have experienced a rapid recovery and reclaimed habitats and haul-out sites in the Baltic Sea and North Sea. While current population estimates of harbour seals (~25,000) [70,71], grey seals (38,000 (counted)) [72,73], ringed seals (>22,800) [74,75,76] and Belt Sea and Baltic Proper harbour porpoises (~17,300 (95% CI = 11,695–25,688; CV = 0.2) and ~500, respectively) [18,77,78,79] from the Baltic Sea, Belt Sea and Kattegat are below the minimum threshold (>250,000–500,000 individuals) for enzootic circulation of morbilliviruses [80], recent population estimates of UK and Wadden Sea grey and harbour seal populations suggest a UK population size of 157,500 (approximate 95% CI 146,00–169,400) and 43,750 (approximate 95% CI 36,000–58,700) [74], respectively, and a Wadden Sea population size of ~39,500 harbour seals [81] and >9000 (counted) grey seals [82]. Meanwhile, the population of North Sea porpoises have been estimated to be ~350,000 (2016: 345,000, CV = 0.18) [83]. Interestingly, genetic studies indicate some recent movements and possible hybridization between the North Sea and Baltic Sea grey seal populations facilitated by the recolonization of the intermediate Kattegat and southwestern Baltic thereby connecting these regions [84]. As such, the total population of the Northeast Atlantic pinnipeds and porpoises could be approaching the minimum threshold for enzootic circulation of PDV and CeMV, opening for future chances in the circulation of these viruses. In addition, recent studies indicate that grey seals may compete with and predate on the resident harbour seals during their expansion to previously populated haul-out sites in North Europe [85,86,87]. As such, the rising number of seals enhances inter- and intraspecies contact rates and promotes pathogen exchange among species and populations. Thus, as the North Sea and Baltic Sea grey seal populations begin to overlap in range, the circulation of infectious diseases may increase in the future as abundance, distribution and contact rates continue to grow.

The 2021–2022 season of highly pathogenic avian influenza (HPAI) accounts for the largest epizootic in Europe observed so far [63]. The persistent circulation of the virus subtypes A(H5N1), A(H5N8), A(H5N5) etc. indicates an endemic state of HPAI H5 (clade 2.3.4.4b) in avian hosts which poses a year-round continuous risk to poultry, humans and wildlife with high-risk periods in the autumn and winter [63]. Spillovers from birds to European pinnipeds have been reported as several isolated cases of H5N8 [43,44,65,66], while the introduction of H5N1 (clade 2.3.4.4b) to New England harbour and grey seals, led to an UME in mid-June 2022 [88]. Despite an increased risk of pathogen exchange between marine mammals of the North Sea and Baltic Sea regions, as well as IAV circulation initiated by spillover from avian reservoirs, current pathogen surveillance efforts in Kattegat and the Southwest Baltic are largely based on the opportunistic sampling of stranded animals with substantial annual variation in the coverage of each species. Only few dedicated blood samples and viral swabs are being collected for serum- and RT-PCR-based pathogen screenings during seal and harbour porpoise captures conducted as a part of satellite tagging studies. However, the addition of samples from live adults, and even more so pups, would greatly help future investigations of viruses and other pathogens. Collection efforts on the sampling of swabs, lung and spleen tissue from necropsied animals in Denmark and Sweden were intensified during the years of 2018 and 2019, resulting in a larger sample size compared to the other years. However, continuous screening of representative sample sets and serological investigations are needed to properly assess the presence of pathogens in the Baltic Sea and North Sea marine mammal populations. As screenings are currently limited to pathogens already known to circulate in the Baltic Sea or North Sea, future monitoring could benefit from metagenomic studies of subsets of carefully selected individuals from each population [89]. Such analyses could help uncover gaps in the current disease surveillance by the simultaneous screening of the animals for a broad range of known and novel pathogens and thereby provide researchers with an extended understanding of the pathogens circulating in the resident marine mammal populations.

## Figures and Tables

**Figure 1 pathogens-12-00357-f001:**
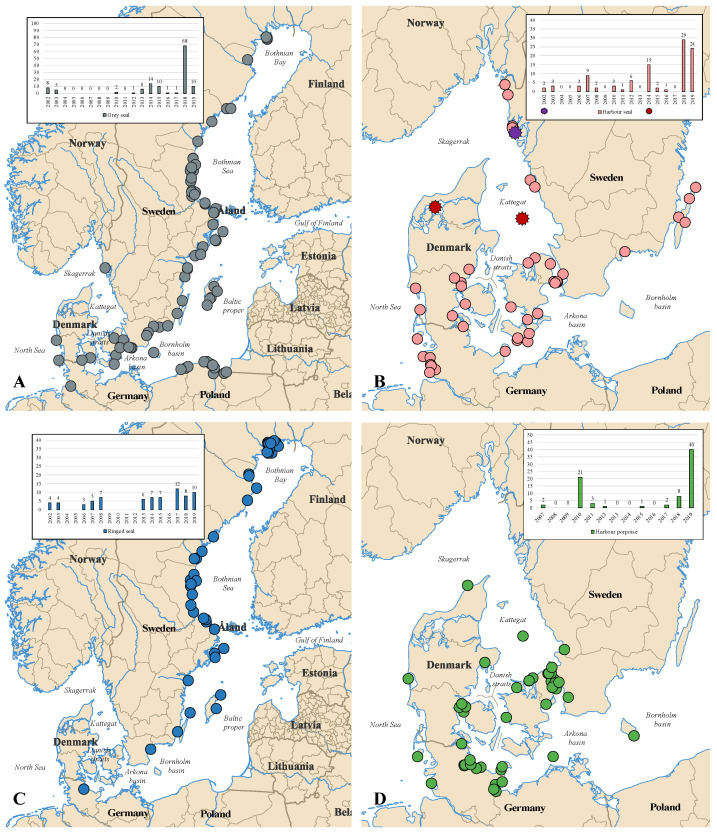
Map of collection sites in the North Sea–Baltic Sea region and a histogram illustrating the distribution of samples screened per year color-coded by species. (**A**) map and histogram of grey seals with grey dots and bars, (**B**) map and histogram of harbour seals with light red dots and bars, (**C**) map and histogram of ringed seals with dark blue dots and bars and (**D**) map and histogram of harbour porpoises with green dots and bars. The small stars indicate the last PDV outbreak (2002) and the last IAV outbreak (2014) as well as collection sites of the positive individuals (PDV = purple and IAV = dark red).

**Table 1 pathogens-12-00357-t001:** Overview of total number of samples based on species, region and year.

	Grey Seal				
**Area**	**2002–2003**	**2006–2008**	**2010–2012**	**2013–2015**	**2017–2019**
Skagerrak/Kattegat/the Sound/Danish straits/	1		3	3	
Arkona Basin	1			2	5
Bornholm Basin	1			5	8
Baltic proper	3			5	27
Bothnian Sea	5			9	34
Bothnian Bay	2			5	2
North Sea				1	6
	Harbour seal				
**Area**	**2002–2003**	**2006–2008**	**2010–2012**	**2013–2015**	**2017–2019**
Skagerrak/Kattegat/the Sound/Danish straits/	3	14	5	9	32
Arkona Basin			4	1	
Bornholm Basin					1
Baltic proper	2		1	2	
Bothnian Sea					
Bothnian Bay					
North Sea				2	19
The Limfjord				4	
	Ringed seal				
**Area**	**2002–2003**	**2006–2008**	**2010–2012**	**2013–2015**	**2017–2019**
Skagerrak/Kattegat/the Sound/Danish straits/					1
Arkona Basin					
Bornholm Basin		1			
Baltic proper	2	1		3	1
Bothnian Sea	5	8		7	3
Bothnian Bay	1	5		10	25
	Harbour porpoise				
**Area**	**2002–2003**	**2006–2008**	**2010–2012**	**2013–2015**	**2017–2019**
Skagerrak/Kattegat/the Sound/Danish straits/		2	24	1	46
Arkona Basin					
Bornholm Basin					1
Baltic proper					
Bothnian Sea					
Bothnian Bay					
North Sea			1		3

**Table 2 pathogens-12-00357-t002:** Overview of the number of screened individuals divided into species and age groups. IAV and PDV represent the number of positive individuals for each virus and IAV % and PDV % represent the percentage of positive samples.

	Harbour Seal								
**Year**	**Pups/calfs**	**Juveniles**	**Adults**	**NA**	**Total**	**IAV**	**PDV**	**IAV %**	**PDV %**
2002	-	1	1	-	2	0	1	0	50
2014	1	5	4	5	15	2	0	13	0
2002–2019	14	34	34	17	99	0	0	2	1
	Grey seal								
**Year**	**Pups/calfs**	**Juveniles**	**Adults**	**NA**	**Total**	**IAV**	**PDV**	**IAV %**	**PDV %**
2002–2019	12	35	51	28	126	0	0	0	0
	Ringed seal								
**Year**	**Pups/calfs**	**Juveniles**	**Adults**	**NA**	**Total**	**IAV**	**PDV**	**IAV %**	**PDV %**
2002–2019	21	8	20	24	73	0	0	0	0
	Harbour porpoise								
**Year**	**Pups/calfs**	**Juveniles**	**Adults**	**NA**	**Total**	**IAV**	**PDV**	**IAV %**	**PDV %**
2007–2019	9	32	11	26	78	0	0	0	0

## Data Availability

All metadata for samples included in the study has been listed in the Appendix A.

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
