# Peer review of "Screening for Influenza and Morbillivirus in Seals and Porpoises in the Baltic and North Sea"

_pathogens, 2023, doi:10.3390/pathogens12030357_

Round 1

Reviewer 1 Report

This is a well-presented paper bringing us up-to-date on the status of two important viral pathogens in the Baltic and North sea seal and porpoise populations. It highlights the limitations of current largely opportunistic sampling efforts and makes the case for further continued surveillance for pathogens in these populations. 

One suggestion for the presentation of results I would make is to make explicit, the ages of the animals that were sampled. Eg. for seals the number of pups, juveniles and adults (or unknown). This information is given in the supplementary information but it would be useful to have brief summary in the paper. It would also add to the discussion on considerations for future surveillance efforts especially given the distinct age profiles of viral vs sero-positivity of influenza found in grey seals elsewhere.

In future screening efforts I would also suggest testing the sensitivity of the M-gene-MGB probe system rRT-PCR for detection of influenza A viruses (Nagy et al 2021 Plos One) in addition to the NP gene primers used here. 

Reviewer 2 Report

The manuscript entitled " Screening for influenza and morbillivirus in seals and porpoises in the Baltic and North Sea" provides new insight into the knowledge of these important pathogens among marine mammals. The topic discussed in the following work is of undoubted interest and relevant as well as the study design is appropriate. The major weakness of the work is the length of some sections (such Introduction and discussion), the number of references (96, an article should have more less 50 references) and the description of obtained results. Only table and figures regarding the sampling are available while a table regarding the percentage of positive animals should be provide. If information related age, sex or other data are available I suggest a risk factor analysis although only 3 samples were positive for IAV and maybe not statistically association will be found. The manuscript contains several grammar errors that need to be checked. The article requires an important revision. As follows, some specific comments.

Abstract: This section is too long, includes a lot of introductive information, which could make the reader confused. At the state of the art the abstract section is inconclusive. Moreover, the results obtained are not clearly described. How many positive samples were detected? Which percentage? Please see the author’s guidelines for abstract section.

Introduction:

Line 44-70: Huge information about the biology of marine mammals and ecosystem. Please reduce this section.

Line 80: The authors mention a few morbilliviruses, as well as phocine distemper and CeMV. I believe it is beneficial to the reader that there is consistency or that it is mentioned that it is the same infection. Furthermore, some information on the diseases mentioned is absolutely lacking (such as mention of etiology and pathogenesis among cetaceans). Information concerning the distribution of these viruses among marine animals could be best spent in the discussion section, where the findings are compared to those of other research.

 Line 136: This reference needs to be better cited.

Line 162: This is not a result. This statement should be included in the discussion section.

Please, provide a contingency table regarding the number of tested and positive sample (including percentage).

Line 172-174: Please, just describe your results, it's not a discussion. The results section should have no references. Findings need to be more clearly described. Are other information such age, gender ecc. available?

Discussion: Very long discussion, dispersive, and above all, inconclusive. The discussion should be set up by comparing one's own results with those obtained from other studies, then specifying the meaning of one's own results, highlighting limitations and possible explanations for positivity and negativity.

Reviewer 3 Report

Authors investigated the presence of influenza A virus and morbilliviruses in tracheal swabs and lung tissue samples from 379 marine mammals collected in the Baltic Sea and North Sea between 2002-2019. No sera samples were tested (not available). Only three harbour seals tested positive for IAV.

Screening for virus in wild animals is very important and publish negative results as well.  I some recommendations and remarks:

Did you sequenced positive IAV samples? Could you include the qPCR Ct values?

What about samples preservation? did you use some medium to preserve the material? Samples 2002-2017  obtained from the archives at the Institute for the Environmental Specimen Bank (ESB) at the Swedish Museum of Natural History were preserved at -20C. Did you quantify the genetic material if it is not degraded? Did you use some internal control for PCRs? What kind of positive controls were used for PCR?

Round 2

Reviewer 2 Report

The authors have responded satisfactorily to the questions I raised. I consider the manuscript ready for publication.